# A Methodology for the Digitalization of the Residential Building Renovation Process through OpenBIM-Based Workflows

Alberto Armijo [1,*] , Peru Elguezabal [1,*] , Natalia Lasarte [1] and Matthias Weise [2]

1 TECNALIA, Basque Research and Technology Alliance (BRTA), Astondo Bidea, Edificio 700, Parque Tecnológico de Bizkaia, 48160 Derio, Spain; natalia.lasarte@tecnalia.com
2 AEC3 Deutschland GmbH, Archivstr. 21, 01097 Dresden, Germany; mw@aec3.de
* Correspondence: alberto.armijo@tecnalia.com (A.A.); peru.elguezabal@tecnalia.com (P.E.)

**Abstract:** The European building industry is facing a strong increase in renovation processes, which are still non-cost-effective, involve unproperly coordinated stakeholders, are disturbing for the occupants, and cause important inefficiencies in the overall renovation process. In this context, digitalization and Building Information Modelling (BIM), as an enabler, is the key solution that may drive renovation interventions to ensure a more successful and leaner process, aiding the whole value chain of actors to achieve its full potential. This research describes the OpenBIM methodology applied in order to transform the implicit knowledge from the stakeholders involved in the building renovation process, not structured enough for automation, into an OpenBIM digital process based on the BIM standards. The outcomes of this research are the OpenBIM ready workflows that represent the renovation process and information requirements according to the involvement of different stakeholders rooted in the analysis of barriers, requirements, and needs. Those workflows are the basis for the future development of specific products and tools for boosting digitalization and interoperability in the renovation process.

**Keywords:** building renovation; BIM; interoperability; OpenBIM; BPMN; workflows; digital construction; retrofitting; IDS

## 1. Introduction

### 1.1. Context and Rationale

The impact of the construction sector on the environment is undeniable. The energy inefficient building stock is responsible for approximately 36% of all $CO_2$ emissions representing 40% of the energy consumed in the Union [1]. Thus, there is a strong necessity to construct new highly efficient buildings, but, at the same time, the actual stock needs to be deeply retrofitted to update all those buildings to actual standards [2]. Consequently, an appropriate policy and awareness for boosting the building renovation are crucial. An improvement of the renovation rate, from the current 1% average to 2% within the next 10 years, will lead to the proposed 55% emission reduction target for the EU [3].

The lifecycle of building construction is often complex [4] due to the huge amount of activities and stakeholders involved. In addition, the lack of standardization makes it rather hard to understand, since it comprises a vast number of typologies, phases, activities, and stakeholders intervening in the process [5], which often requires different strategies to tackle the issue. As a result, current construction and renovation processes are highly inefficient [6], making construction one of the most unproductive sectors [7]. Although technological advances are a great help in the effort to solve or mitigate the consequences of these problems, at present, no appropriate solutions have been found.

In this context, Building Information Modeling (BIM) [8] is adopted as a very promising methodology to improve the whole building process [9], for new construction [10] as

well as for renovation works [11]. Based on the application of collaboration principles, the adoption of BIM in the construction sector can significantly improve the efficiency of the overall process [12–14], with a direct impact on the reduction of costs, time, and waste.

The strategy using BIM is mainly based on obtaining a common framework to work together by means of conceiving digital-based models that are to be shared among partners. Its main goal is that the involved stakeholders develop, in collaboration, a digital representation of the building, a design that is continuously updated, reaching its highest level of development once a digital twin is generated [15,16]. This design provides reliable information about the different elements and components as well as the actuations planned. The exchanged data include a combination of geometric and semantic information, and the centralized models allow one to easily extract and exchange this information. The latest version of the model is accessible to all the participants and communication channels are established, thus cooperation is promoted, and uncertainties are reduced [17]. In recent years, BIM-based software has significantly increased its presence in applications related to architecture, engineering, and construction (AEC), covering all the phases and types of applications [5,9,17–21]. Its application worldwide in different countries with various working cultures and economies has also been studied in the past [13,14,22–25].

One of the main characteristics of this industry is the concurrence of multiple and different actors, with various skills, knowledge, tools, applications, and software. All that information needs to be generated, adapted, exchanged, and interpreted between them. To make the process and the overall result of the actuation more efficient, a synchronization of this information is needed based on interoperability principles (i.e., how to connect systems and applications on a technical level) [26]. In parallel, and as part of this information exchange, communication needs to be facilitated between the participants. The current digital revolution with the proliferation of big data and AI should facilitate this change of paradigm [27].

The BIM methodology relies on interoperability as the way to reduce the barriers that hinder the exchange of information, as a result of the use of open standards. Among the possibilities for solving interoperability impediments that represent a barrier to information flow, OpenBIM standards such as IFC (ISO 16739) [28] and related standards are being promoted by buildingSMART. Moreover, related to this interoperability issue, the structuring and systematization of information lead to efficient construction projects, by means of arranging the collaboration workflows. Interoperability is a wide concept beyond the technical software development aspect, and the approach is also based on organizational and procedural domains, a concept that also requires a cultural change [29]. Collaborative business processes promoted by interoperability represent the path to overcoming traditional ways of interacting, a set of rules and principles that may hinder technical connections between stakeholders.

However, the application of BIM is neither easy nor straightforward. First, several barriers are currently present in the sector [5,30–32] that do not stimulate higher levels of BIM in the renovation activities [9,33] and, second, the application of BIM to projects and companies is not an all-or-nothing concept, as different levels of BIM can be considered [34,35], and even a BIM maturity level is suggested by some authors [36]. The promotion and significant increase in the use of BIM in the building industry is still a very ambitious challenge that requires deep and important changes in current working cultures. Different methods are feasible in order to reach the final goal of higher rates of BIM application in the sector, and currently, several research initiatives are working in that direction.

The research presented in this paper aims to contribute to higher rates of application of the BIM methodology in the renovation sector, by means of standardization of digital-ready workflows. These will represent the renovation process and information requirements in a structured framework, according to the involvement of different stakeholders. This information will provide the basis for future development, as part of the BIM4ren project [37], of tailormade products and tools to be further developed and solutions that will also

be defined to overcome the barriers identified for higher penetration of BIM technology. Details regarding the specific goal of the research are presented in Section 1.3, following the contextualization of this research as part of the BIM4Ren project under Section 1.2.

*1.2. BIM4Ren Project Background*

This paper is based on research that is being developed through the European research project, BIM4Ren [37], which involves a total of 23 European partners (11 SMEs, 2 Large Enterprises, 7 Research Centers and Universities, and 3 National or European Associations) spread across 10 countries. The main target of the project is to enhance the efficiency of the renovation sector trough digitalisation. Consequently, digital tools and a common platform are being developed, oriented to streamline the renovation process according to the user's needs. These tools are being iteratively tested and validated in three different pilot sites in Europe (Spain, France, and Italy), either real projects as pilots or demonstrators of the findings of the project.

In this project, the end user's involvement in the discussion is key to ensuring that the developments are user-centered and focused on real needs and requirements. For that purpose, a research method called Living Labs [38–40], based on an open innovation approach for sensing, prototyping, validating, and refining complex solutions in multiple evolving real-life contexts [41], has been implemented as part of the strategy adopted by the BIM4Ren project. Within this Living Lab framework, activities intended to integrate the users' insight and expectations in decision making have been conducted. The Living Lab developed in this case has adopted a process that combines surveys, interviews, and workshops supported by literature reviews [42].

The main goal of the BIM4Ren project is to develop a set of digital tools, tailored and adapted to the specific needs of each typology of users. The set of technologies developed will be orchestrated through a BIM-based One-Stop Access Platform (OSAP) that will contribute to facilitating the communication and information exchange between the tools composing the overall system. The research activities consist of a three-step approach. First, an elicitation process about the difficulties, requirements, and expectations from different stakeholder groups was conducted. Then, the information collected in that initial step was the basis for the systematization of activities and processes, leading to the development, of the digital workflows in the second step [43]. These workflows will establish the underlaying schema for the digital tools to enforce interoperability. The final step will consist of the development of those tools together with the OSAP, and the testing and validation of the platform. The Living Labs and the pilot sites accompany the whole process and are used as a benchmark to compare the main achievements obtained in those phases.

The first step, the elicitation process, has already been conducted [42]. The result was a total of 229 surveys answered by European stakeholders, including all the relevant actors in the value chain, 19 interviews with agents working directly on the pilot case projects, as well as 5 workshops held around the pilot sites. The workshops represent the highest level of development for the Living Lab, where the results achieved from the surveys and interviews are further elaborated and developed through discussions among involved stakeholders, with a special interest in the different perspectives and points of view of the main actors. There is a twofold outcome of the elicitation process:

- The barriers and difficulties for the adoption of BIM were discussed and individualized for every type of stakeholder category. The main concerns relate to time savings, cost savings, and collaboration processes, which are generic for any construction process. Retrofitting has specific hindrances related to gathering the data of the building's characterization parameters.
- In parallel, renovation processes have been mapped, distinguishing different typologies of stakeholders and renovation works. This discussion also allowed for the extraction of significant information reflecting the As-Is situation. The main processes

and the information exchange between domain experts, especially in the detailed design and later phases, were collected.

Based on those two main outcomes, the conclusion is that, on one side, the interoperability of software solutions is considered a key element to progress towards higher levels of cooperation and BIM application levels. On the other side, the interaction between participants and the information exchange process needs to be systematized and structured.

### 1.3. Objective and Scope of This Research

The content presented in this paper is framed in the second step of the research developed in BIM4Ren, as presented in Section 1.2, where the outcomes of the first step of elicitation are employed to structure the information and systematize processes as the basis to develop the tools in the third step. Therefore, the main interest at this stage is to generate tailored digital workflows [43] for different stakeholders, workflows that will be later used as the basis for the development of digital solutions. This comprises the application of a methodology to understand the implicit connections among stakeholders during different phases of the process, as well as to identify the different exchanges of information between those participants in each stage.

To substantiate the research presented in this paper, the outcomes of the elicitation process around the Living Labs were considered. Nevertheless, it is necessary to highlight that the methodology and procedure can be used as a reference for application in other potential use cases in renovation activities.

The methodology applied to transform the knowledge from the stakeholders into a digital process is rooted in the OpenBIM methodology [44], as presented in Section 2. By means of an example of a façade renovation case in Section 3, a specific description of the application of the methodology is provided, increasing the level of detail of the information that is exchanged in the different steps. Finally, the main conclusions regarding the potential use of this methodology together with the interest generated about the workflows are discussed in Section 4, where future action in the ongoing research is also anticipated.

## 2. Methodology for Defining a Workflow

The methodology presented in this Section relies on several standards promoted by BuildingSMART International (BSI) [45], which is the industry body that boosts standardization in BIM as a meaningful driver of the digital transformation of the construction sector. It is aimed at promoting the use and dissemination of open data standards, collaborative processes, and integrated practices in the building and construction industry, enabling digital transformation. It promotes the international consensus among stakeholders using specific standards divided into the following categories, depending on their purpose. The main standards are enumerated below:

- IFC (Industry Foundation Classes) is the specific data model to exchange information of the model through different software.
- IDM (Information Delivery Manual) is the methodology for defining and documenting processes and data requirements. It is usually complemented by the graphic language BPMN (Business Process Modelling Notation), which provides graphic notation to understand and represent the communication between participants and processes.
- MVD (Model View Definition) is the subset of the IFC scheme aimed at supporting a set of data exchange requirements. The concept of MVD was recently extended by IDS, the Information Delivery Specification, which focuses on an easy way to formalize information needs and its representation in the IFC format.
- IFD (International Framework for Dictionaries) is an international dictionary whose purpose is to uniquely clarify the definitions and meanings of entities, products, and processes in the construction industry.

To conceive the present methodology and illustrate the collaborative rehabilitation processes and information flows using natural language, some of the above-presented standards, i.e., those that provide more value to this methodology elaboration, were taken

into consideration and adapted from the OpenBIM [44] paradigm. Namely, the leveraged standards are IDM, MVD, and IFD.

On the one hand, and as a first step of the present methodology, the IDM (Information Delivery Manual) standard was applied. Based on the characterization of the information of the renovation process, presented in Section 2.1, the main advantage that the IDM offers is the precise definition of which user, and at which specific point in time, must provide and combine the required information. This information exchange was formulated as Exchange Information Requirements (EIR), which are rooted in the high-level data to be exchanged during the process. The IDM was complemented by the graphic language BPMN (Business Process Modelling Notation) [46], as described in Section 2.2. BPMN provides an interoperable and standard representation to understand and represent the communication and data exchanges between participants and activities. The use of this standard is crucial to reach a consensus between technical and non-technical actors regarding the information flow between them since it provides a structured way to graphically share processes. Once agreed upon, the workflows can be deployed in software systems that provide process management [47,48] capabilities. These software systems, commonly known as workflow engines, support continuous improvement functionalities with respect to the process performance measurement, i.e., tracking the time spent in each of the phases within the workflows.

On the other hand, and as the second step of the present methodology, the Model View Definition (MVD) was used to model the processes and the high-level information exchange requirements, as explained in Section 2.3. Since the IDM does not offer low-level information or guidelines on the technology that should be used to exchange the information from the building models, the MVD approach was considered to define a subset of the IFC schema required to satisfy one or many of the identified EIR. MVD requires specific implementations by software vendors to enable interoperability. Soon, the new standard version IFC5 will be modular [49], with a shared base, and the definition of exchange requirements will be carried out using Information Delivery Specifications (IDS). However, the approach following the IDS will be compatible with the methodology presented in this article.

Regarding the IFD standard, this was leveraged for unifying global terminology and supporting the elaboration of the present methodology. The goal of this standard is to provide a common library of technical languages that improve the collaboration between construction actors aiming at smooth interoperability and consistent information exchange regardless of the language used.

In summary, the methodology proposed for capturing and generalizing a building renovation workflow [50] comprises two steps. These steps are interrelated and recommend the use of mutually agreed upon techniques to identify and drill down through the pieces of data, information, and knowledge required to enable seamless organizational, semantic, and technical interoperability capabilities during collaboration within the building renovation process. The right side of Figure 1 shows the methodology concept elaborated in this work from the point of view of the main components involved and their relations. The left side of Figure 1 shows the building blocks that comprise the OpenBIM standards paradigm.

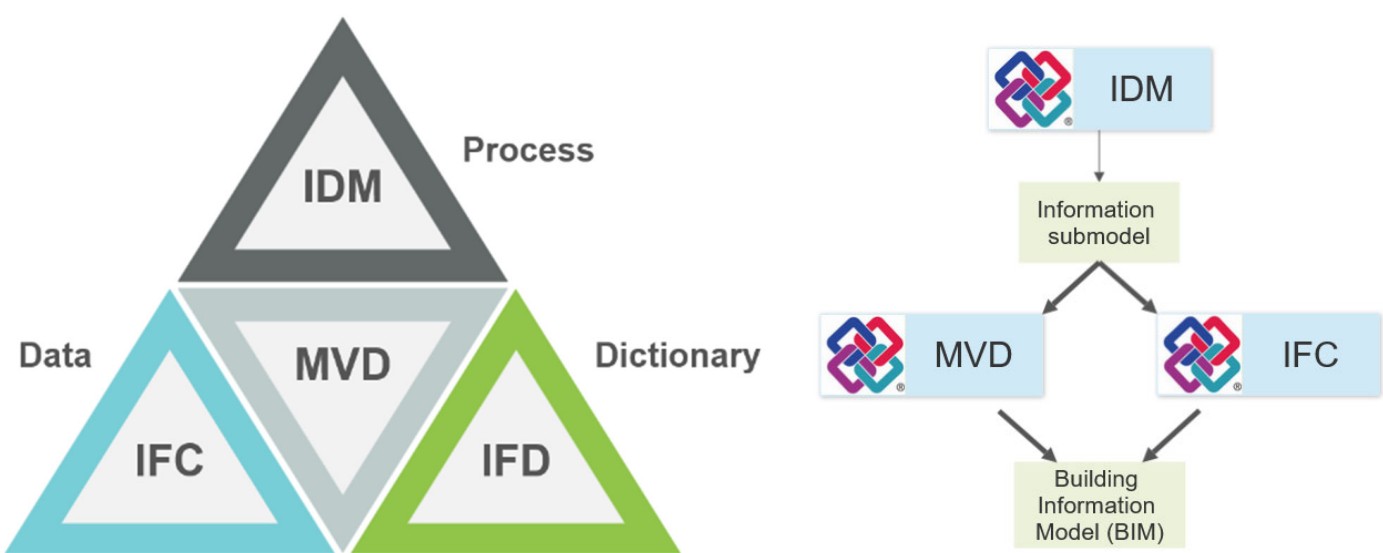

**Figure 1.** Methodology concept. Main components involved (**left**) and application sequence (**right**).

### 2.1. Characterization of the Information of the Renovation Process

The main objective of the characterization of the information is to set a commonly understood reference framework of the renovation process as it is, to consider the classification in the definition of workflows and the definition of use cases. Particularly, a taxonomy of the renovation process in terms of typologies, stakeholders, and phases was defined, which will pave the way for defining the overall workflow in renovation in a standard way.

To characterize the type of renovation intervention proposed, some authors have considered the different technologies to be adopted [51,52] while others focus on the final target of the intervention [53]. A combination of both the technology together with the specific result pursued after the intervention can also be considered [54,55]. In the BIM4Ren project, a categorization of renovation typologies was carried out that was further deepened through the elicitations gathered from the stakeholders participating in the Living Labs [42]. The standardization of the typologies, as well as the type of stakeholders involved and the tasks performed in each one, enabled the systematization of workflows. The typologies cover a range of factors, from the single works prior to renovation, such as maintenance and inspection, to more complex processes, such as deep renovation. In this article, to exemplify the definition of workflows, the typology identified as "renovation in the envelope" was selected to be addressed.

On the other hand, considering the difficulties for cooperation in such a complex environment, it is becoming essential to automate the workflow with an appropriate identification of phases and activities, where the interrelations are well defined. Although many efforts have been conducted for the characterization of the construction phases according to several standards (UNICLASS2, OMNICLASS, PAS 1192) [56–58] and renowned associations, such as, RIBA [59], AIA [60], NATSPEC [61], CIC [62], etc., there is no agreed-upon standardization of the renovation phases. The renovation characterization must take into consideration the singularities of the renovation, which are mainly focused on the interaction of the works with the existing building and the involvement of the tenants, whose activity must be made compatible with the refurbishment works.

On the basis of one of the most common classifications, i.e., the RIBA plan of work [59] for construction stages, the phases of a renovation process were set as follows, considering the particular approach of the retrofitting [63,64]:

1. Strategic definition.
2. Information gathering and survey.
3. Diagnosis.

4. Renovation conceptual design.
5. Renovation technical project.
6. Construction.
7. Handover and close out.
8. In use.

Phases such as "information gathering and survey" and "diagnosis", which are the cornerstones of the renovation, were added to the standard classification. The first one is devoted to collecting information about the existing building such as geometry, materials, elements, constructive sections, pathologies, etc. It involves surveys and visual inspections, destructive or non-destructive techniques for the characterization of elements and/or materials, as well as technologies for data capture such as photogrammetry or 3D scanners. The second one consists of the assessment of key parameters of the existing building to set the most suitable solution for retrofitting according to the customer's requirements and the state of the building.

The analysis of the stakeholders in a construction process is not the target of this paper, but rather to harmonize them in a framework with the common types of stakeholders involved in a renovation process. The categorization of stakeholders and their roles and responsibilities, according to the outcomes of the BIM4Ren project [42], supported by the approach suggested by other authors [65–67], are described below:

- Designer is responsible for the design and consolidation of designs. The role of the main designer can lie with the Architect or Project Manager.
- Contractor, responsible for construction activities and the implementation of designs.
- Sub-contractor works for the main contractor and responsible for discrete and independent functions, such as HVAC, structure, foundations, electricity, plumbing, etc. Particularly, in a renovation project, the survey and data gathering, i.e., 3D Scanning, characterization of materials, inspections, etc., can also be considered sub-contracting activities.
- Residents are actors that set the requirements regarding the project and must be considered in the on-site work planning, as they are the parties most affected by the work.
- Building owner is responsible for the final decision making, defining the purpose of the project, and the end users' constraints. They play several roles depending on the type of property (public, private) or dependencies (owners, social housing, tenants, facility manager, etc.).
- External consultants, such as the software developer or BIM consultants, support the design team through the specific use of tools or software for the purpose required.
- Supplier/manufacturer is responsible for supplying material and equipment to the specifications defined in the project.
- External certification is responsible for validating the project regarding different criteria set by the end users. They can provide standard labels for the final product or the process.
- Public administration is a regulator entity, such as the government, local or security authority, waste managers, representatives of local and public authorities, etc. They supervise, and may set constraints for, the project execution in terms of specific domains, ultimately approving the project.
- Financial entities, or funders, act as the sponsor of the project, funding the budget. They usually have no requirements or personal interest in the project.

### 2.2. Step 1: Definition of IDM OpenBIM Worflows through BPMN

The IDM OpenBIM standard was considered to digitalize and systematize the building renovation process, as presented on the right side of Figure 1. In this regard, to streamline the digitalization of the renovation process, the Business Process Modelling Notation (BPMN 2.0) was leveraged for modelling the envisaged Use Cases. The BPMN [68] is a standard from the Object Management Group (OMG) that promotes the modelling of business processes. It provides a set of graphical elements for describing business processes

based on flowcharts that resemble the activity diagrams from the Unified Modelling Language (UML). The BPMN Version 2.0 of the standard was developed in 2010, and the current revision of the specification was published in 2013.The BPMN is meant to bring together business process design and process implementation, by providing a standard model executable in software workflow engines [69] and standard notation. This notation can be easily understood by all stakeholders, where business analysts (e.g., designers) are devoted to creating and refining the processes, technical developers (e.g., contractors, suppliers, and BIM technical experts) to the implementation, and business managers (e.g., tenants) to monitoring the business processes. These processes can be ideally driven with the support of workflow engines.

The BPMN models are comprised of diagrams built from a limited set of graphical elements. They simplify the understanding of business activities and processes for both business users and technical users. The five basic BPMN element categories are [43]:

- Flow objects: These are the main elements consisting of three core elements: Events, activities, and gateways (see Figure 2). An event is represented by a circle and denotes something that happens, compared with an activity, which is something that is to be carried out. As such, an activity is represented by a rounded-corner rectangle and describes the kind of work that must be conducted. An activity is a generic term for work that a company performs, and it can be atomic or compound, manual or automated. To finish, a gateway is represented by a diamond shape and determines the forking and merging of paths, depending on the workflow conditions expressed as variables.
- Connecting objects: These are used to connect flow objects to each other. There are three types of connecting objects: Sequences, messages, and associations (see Figure 3).
- Swim lanes: These are placeholders for organizing activities. There are two types of swim lanes: Pools and lanes (see Figure 4). A pool represents major participants in a process, separating different actors. A pool contains one or more lanes, which are used to categorize activities within a pool according to the function or role.
- Artifacts: These are used to provide additional information about the process, as seen in Figure 5. There are two standardized artifacts, but modelers or modeling tools are free to add as many artifacts as necessary.
- Data: This is represented with the four elements data objects, data inputs, data outputs, and data stores (see Figure 6). Data objects provide information about what activities are required to be performed and/or what they produce, and can represent a singular object or a collection of objects. Data input and data output provide the same information for processes. Thus, data objects show the reader which data are required or produced in an activity or task, such as the EIR needed to enable efficient exchange of information, i.e., interoperability capabilities, between stakeholders during the collaborative process.

Start

Intermediate

End

Exclusive

Event-Based

Parallel
Event-Based

Inclusive

Complex

Parallel

or X

Task
Name

Sub-Process
Name

**Figure 2.** BPMN events (**left**), gateways (**right**), and tasks (**under**).

Sequence Flow          Message Flow          Association

**Figure 3.** BPMN connectors.

Name

Name
Name

Name
Name

**Figure 4.** BPMN pool and lanes.

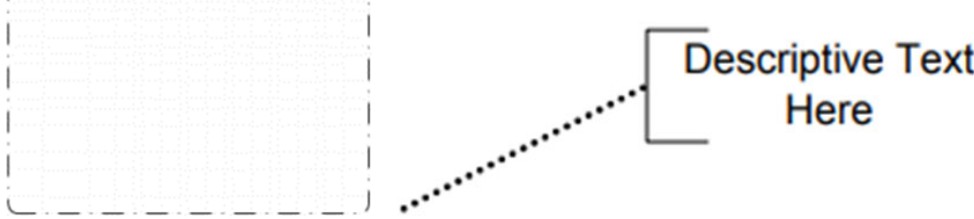

Descriptive Text
Here

**Figure 5.** BPMN group and text annotation artifacts.

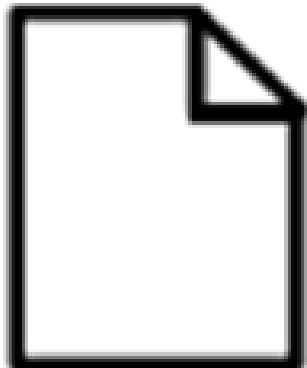

**Figure 6.** BPMN data object.

Accordingly, the creation of the renovation workflows was accelerated by the leverage of these standard building blocks. To foster collaboration, it is necessary that all the participants in a process are aware of the information requirements in terms of correct timing and content. In this regard, the IDM methodology supports the capture and description of the processes and information flows during the life cycle of a building (ISO 29481-1) [70]. The standard itself simply describes the method to produce a text manual for describing the information exchange. However, buildingSMART promotes the use of BPMN graphical notation as a companion to IDM to accelerate the characterization and understanding of the process by all the stakeholders. Consequently, the main benefit attached to BPMN is that it allows one to define which actor, and at which specific point in time, must provide and collate the required high-level information, i.e., the required data formulated as EIR. The EIR is normally listed in a tabular format.

Each renovation work typology (e.g., façade renovation, structural renovation) is considered as a use case. From the software engineering perspective, a use case is a list of actions that define the interactions between an actor and a system to achieve a goal or requirement. To elaborate the BPMN workflows based on these use cases, the outcomes from the elicitation process as described in Section 1.2 were considered as inputs.

### 2.3. Step 2: Using of MVD for Structuring BIM Requirements

The MVD standard was applied after the implementation of the IDM standard, as presented on the right of Figure 1. IDM was used to model the processes and the high-level information exchange requirements, as explained in Section 2.2. On top of that, the MVD approach was essentially considered to determine a subset of the IFC schema that was needed to fulfill the identified EIR.

After finalizing the definition of the high-level EIR, the current step foresaw the detailing and structuring of the collected information, for instance with the help of the BIMQ tool [71] as displayed in Figure 7. BIMQ follows the IDM/MVD methodology from buildingSMART and thus brings all pieces together, namely the actors, phases, use cases, exchange requirements (broken down into object and properties) and, last but not least, its representation in the OpenBIM IFC standard or, if needed, to other data formats. Figure 7 shows, for instance, the definition of Key Performance Indicators (KPIs) [72], a special type of property that is used to compare and evaluate renovation scenarios. Those KPIs include a mapping definition of IFC, in that case, to user-defined properties, and are then assigned to elements and domain models representing the EIR, as discussed with domain or discipline experts. This essentially means to extract detailed and checkable information from the EIR-tables of the domain experts, who normally do not know the details about how to encode such requirements in a data structure such as IFC. This step follows simple rules and differentiates four main categories:

- Discipline models such as an architectural model, structural model, plumbing model, or electrical model. Such models group a set of information that is typically linked to a specific type of actor who is responsible for delivering the data.
- Element types such as a wall, slab, window, boiler, etc., typically with a position and geometrical representation. Besides being classified according to well-known structures, they can be characterized by further parameters carrying important design information, properties, and geometrical detailing.
- Properties of elements such as name, fire rating, thermal transmittance, etc. They represent a piece of information that is typically defined by a single-value property but could also play a more complex part in the data structure such as a relationship, a time-series, or other structured data.
- Geometry of elements, which is basically a description of the expected level of geometrical detail. This should include information such as that specified in CEN 17412 (LOIN standard) [73], such as dimensionality, appearance, parametric behavior, etc. These data will help to conduct further evaluations, such as clash detection, visualization, calculation of quantities, etc.
- Accordingly, such specifications require a joint effort between domain experts and technical experts who know all details about a particular data format such as IFC, gbXML, or others that are of interest to be used for requested data exchange.

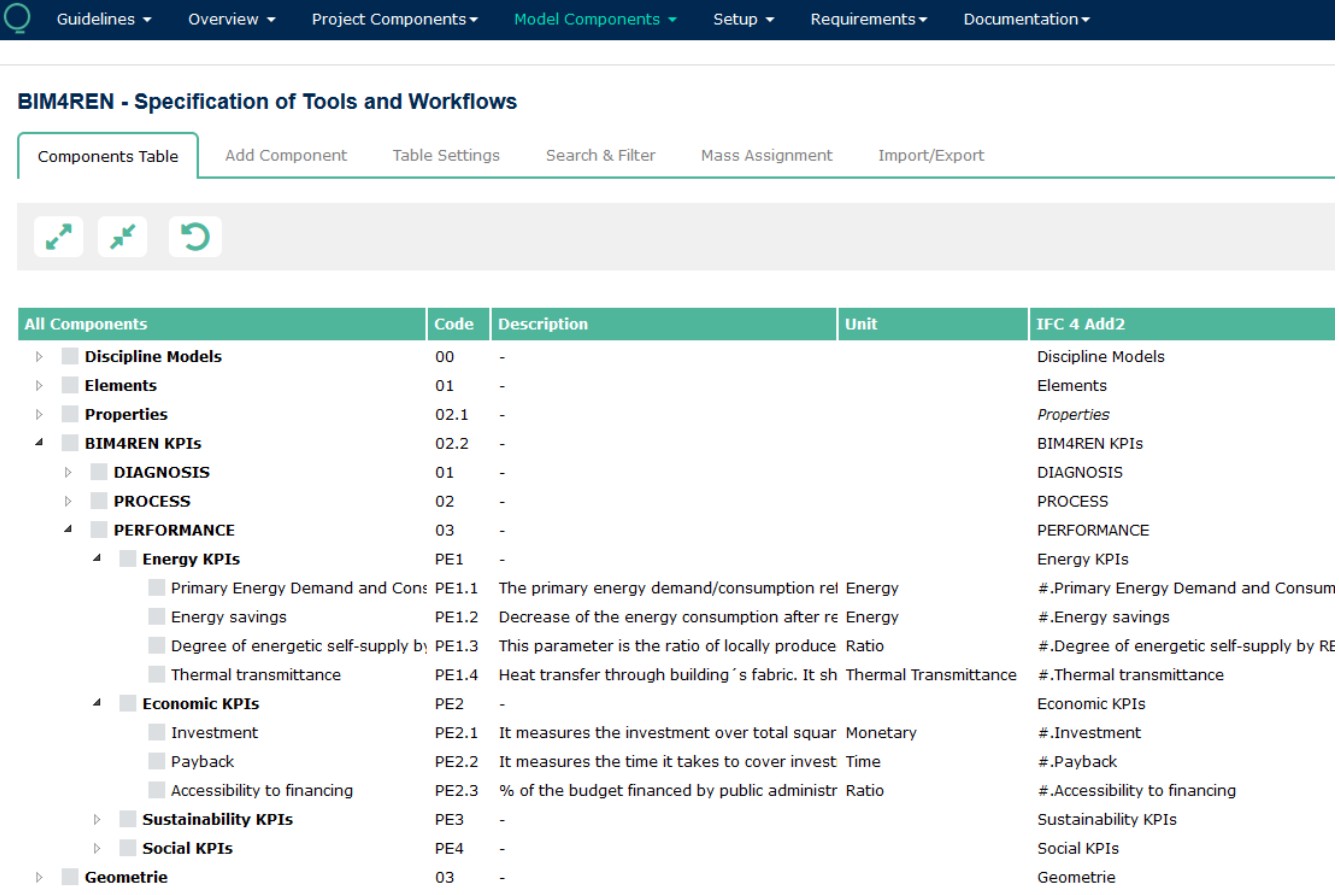

**Figure 7.** Screenshot from BIMQ showing the structuring and detailing of EIRs.

In summary, the IDM and MVD/IFC View Definition were adopted to map the information requirements to the relevant IFC schema elements targeted at streamlining the renovation process. This exercise enables the codification of these requirements into

the software tools that will realize the information exchange, enforcing the BIM technical interoperability perspective.

### 3. Application of the Methodology to a Façade Renovation Use Case

The described methodology was applied in a real use case to showcase, in a graphical manner, the process of a façade renovation. The information regarding the process, stakeholders, and information delivery was gathered from the Living Labs activities held in the framework of the BIM4ren project with the stakeholders involved in the process.

A two-step approach was leveraged to capture the renovation process, first by means of the application of the IDM standard to digitalize the process, representing the process in BMPN diagrams (as per Section 2.2), and second, applying the MVD approach (as per Section 2.3) to provide low-level information about the information to be exchanged between models that enables digitalization.

As a result of the application of the methodology, several workflows of the façade renovation process were conceived from the different perspectives of the agents in the value chain, involved in different stages of the process. The workflows were drawn through BPMN diagrams presented in the enclosed material. Table 1 shows the list of workflows represented in the diagrams for the use case related to façade renovation, as an example of the application of the methodology.

**Table 1.** List of façade renovation workflows created from the methodology.

| Workflow Perspective | Source | Use Case Type |
|---|---|---|
| ID1. Global process | Own research | General diagram |
| ID2. Architect | Interviews | Specific diagram |
| ID3. Private owner | Interviews | Specific diagram |
| ID4. Public owner managing social housing | Interviews | Specific diagram |
| ID5. Contractor | Interviews | Specific diagram |

The "ID1. Global process" workflow represents the lifecycle phases of a generic renovation process from a global perspective. This diagram covers the full retrofitting lifecycle, from the strategic decision to the hand-over and maintenance phases. Figure 8 shows a general diagram of a renovation process as an example of the methodology application, according to the BPMN representation. The first rows represent the tasks of the renovation process from the perspective of the agents involved (owner, designer, contractor, and regulatory entities) whereas the last row is the information exchanged in each phase by typology of the stakeholder. The interaction between phases, stakeholders, and information exchanged is graphically mapped, as it is shown in Figure 8.

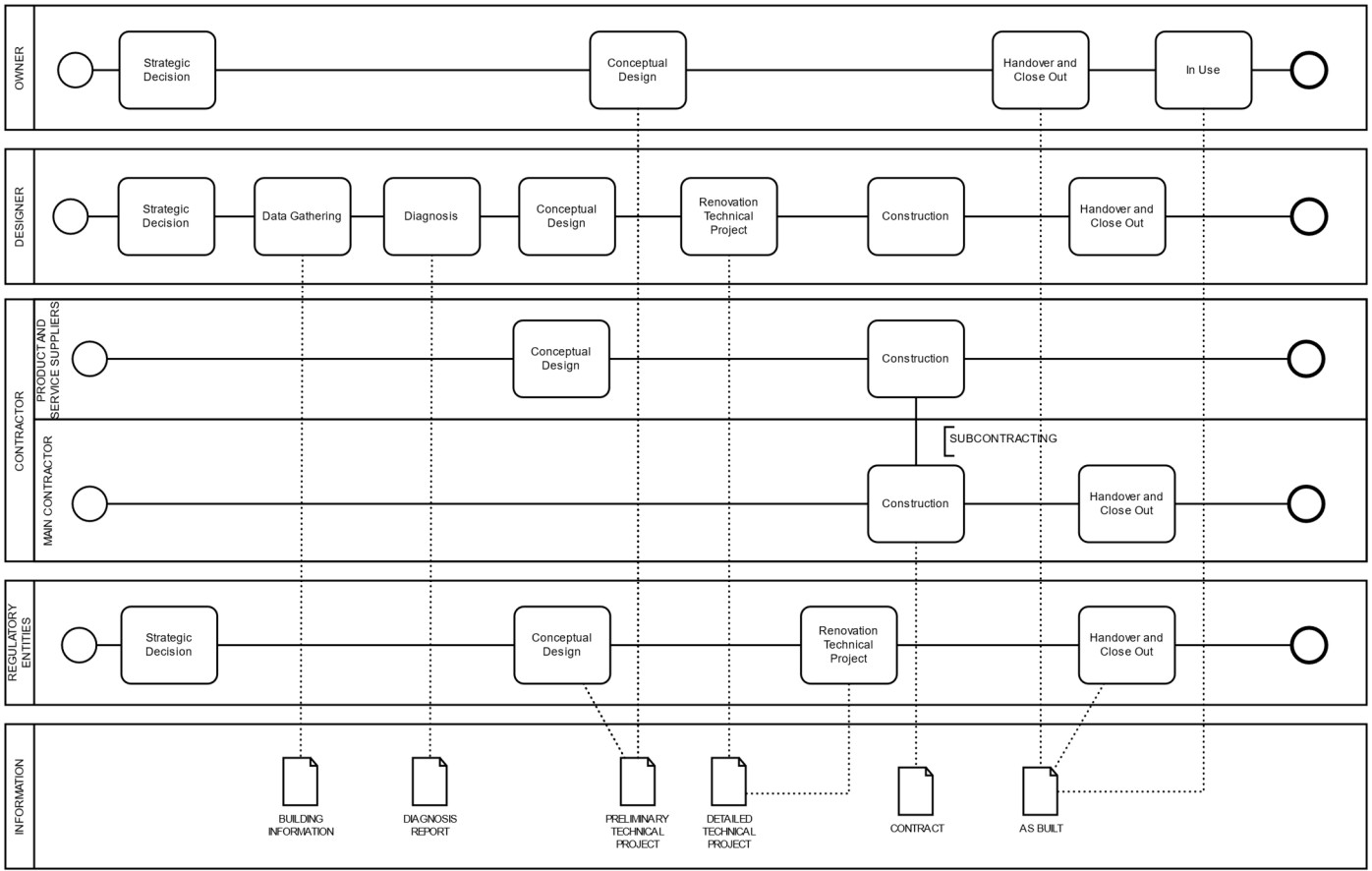

**Figure 8.** Generic renovation process represented in a BPMN diagram (ID1).

The other BPMN workflows (ID2, ID3, ID4, and ID5) listed in Table 1, delivered as companion material to this reading, address the tasks involved in the life cycle of the building more deeply, from the perspective of different stakeholders. As an example, Figure 9 represents the "ID2. Architect" specific diagram. Each task, which represents activity in the generic process (see Figure 8), is expanded with the subtasks and EIRs from the point of view of the architect working in the façade renovation. To enable the comprehension of the "process nesting" in a more visual way, the general diagram including all the phases is presented in Figure 9. Additionally, the "conceptual design" has been amplified to show the subtasks, stakeholders involved, and the exchanged information. The full details of the diagrams are available in the mentioned companion material.

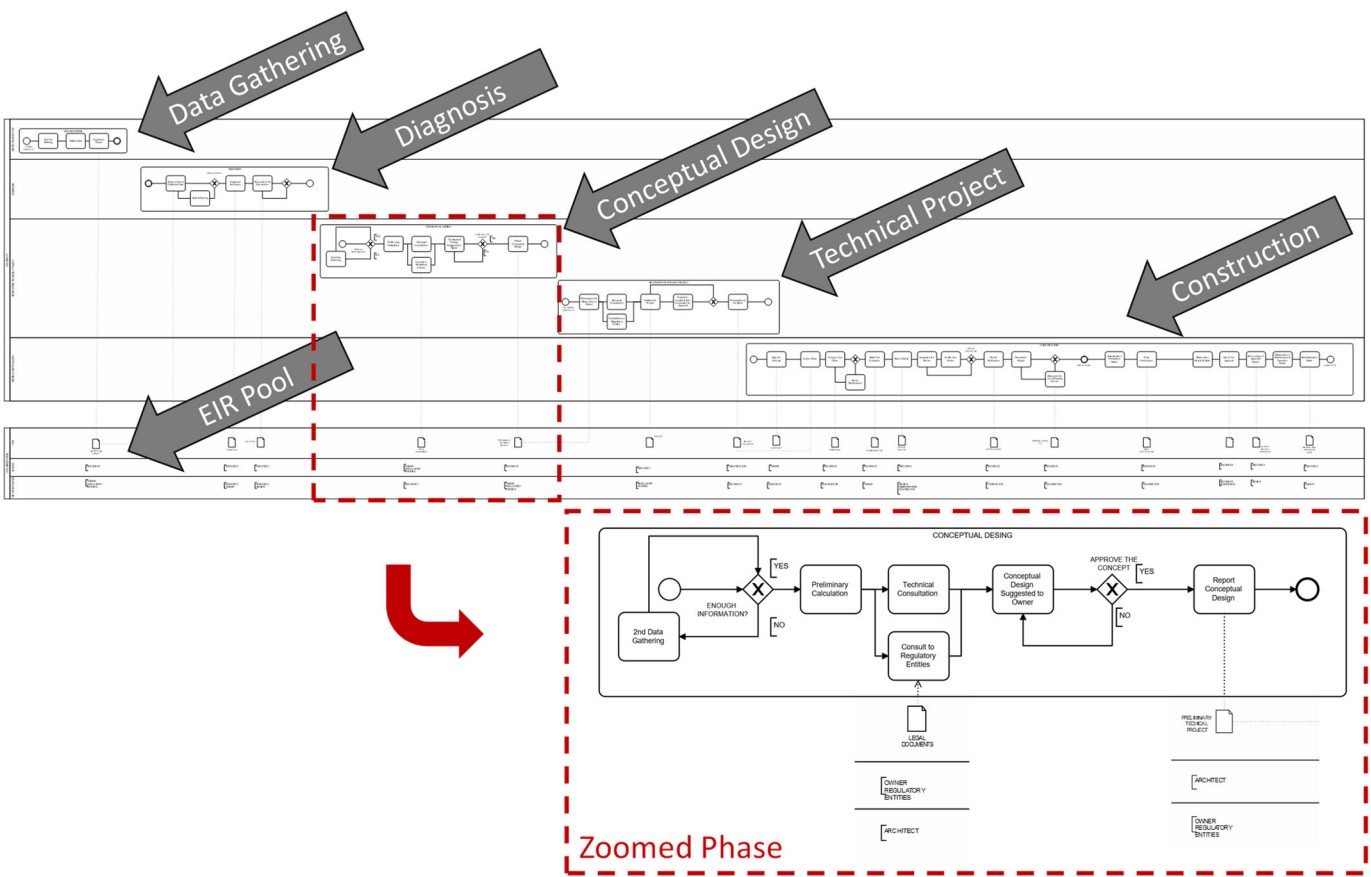

**Figure 9.** Specific façade renovation BPMN diagram (ID2, architect). The conceptual design phase is zoomed.

A transformation of the use cases into BPMN 2.0 workflows is achieved here. The sequence flows, indicating the orchestration between manual and automated tasks (services) are provided. The sequence flows are represented by a solid black line with a black arrow. The manual tasks involve human interaction, while automated tasks involve communication with different software systems. The EIR structure was based on the results from [74] and was collected for each use case in EIR tables (see Table 2), containing basic information. Because of the further elaboration of the diagram shown in Figure 9, through consensus with the involved stakeholders, a validated process as well as the definition of the information exchange between stakeholders is achieved. Figure 10 represents the validated process as well as the consolidated information exchanges between stakeholders at the conceptual design phase.

**Table 2.** Example of the EIR table for the technical Preliminary Energy Simulation Report (4.EIR 2A) from the conceptual design phase.

| Exchange Name | 4.EIR 2A TECHNICAL REPORT: Preliminary Energy Simulation |
|---|---|
| BPMN phase involved | Conceptual Design |
| BPMN tasks involved | # Preliminary Energy Simulations/Dynamic thermal simulation, Simulation of occupant activities, generation of 2nd level space boundaries, converting IFCmodel to gbXML |
| External Data (ED) | - |
| Sending Actor | Engineer |
| Receiving Actor(s) | Architect |
| Possible Tools | IES VE2018, SMACH |

**Table 2.** *Cont.*

| Exchange Name | 4.EIR 2A TECHNICAL REPORT: Preliminary Energy Simulation |
|---|---|
| Description Exchanged Data | - Description of input:<br>- Project Information, Building type, function, orientation<br>- Climate data<br>- HVAC systems<br>- User Types, Number of occupants, Occupants' schedule<br>- Construction and materials properties, U-Value, g-value<br>- Room data; Geometric attributes, dimensions, wall surfaces, room volumes, glazing<br>- Description of output:<br>- Requirements on Energy Efficiency (matching) described as energy targets (e.g., sustainable, environmental, comfort)<br>- Use of Renewable Energy Source (%)<br>- Primary energy consumption for heating, cooling and process ($kWh/m^2$)<br>- Cooling results: cooling demand, cooling demand per $m^2$, cooling load, cooling load per $m^2$<br>- Heating results: heating demand, heating demand per $m^2$, heating load, heating load per $m^2$<br>- Annual $CO_2$ reduction potential (coal, wood, oil, gas, district heat, electricity)<br>- Daylight results: solar radiation from window, Annual solar transmittance<br>- Internal load (people, equipment)<br>- Comfort analysis<br>- Recommendations and strategies<br>- Cost estimation<br>- Conformance to owner requirements |
| Exchange Models<br>Data Exchange | Simulation graphs, report<br>gbXML, XLS, text/csv etc., IFC, PDF |

The definition of the EIR for the selected tasks is demonstrated in the following example of the EIR table (Table 2) for the technical Preliminary Energy Simulation Report (4.EIR 2A) from the conceptual design phase.

Table 2 defines input and output information linked to the use case primary energy simulation, which in the next step must be translated into a data specification for IFC or any other data structure identified as an applicable data exchange model. The focus was to apply OpenBIM standards so that IFC was selected as the main solution for cross-domain information exchange. Accordingly, those requirements must be translated into an IFC-based structure that is essentially able to (1) identify the object type or IFC entity that should carry the information and (2) to describe how this information is represented in the IFC data structure. The use case shown in Table 2 requires, for instance, occupant information to derive thermal loads and further temperature requirements from the occupant schedule. This information is typically attached to a room, which in IFC is represented by an IfcSpace entity. Room-related occupant information can be defined using the Pset_SpaceOccupancyRequirements, including properties such as OccupancyType, OccupancyNumberPeak, or OccupancyTimePerDay.

The whole process for structuring EIRs and working on mapping specifications is supported by the BIMQ tool from AEC3. Figure 11 shows the definition of IFC mappings for identified KPIs attached to the building, represented by an IfcBuilding entity. The user can either select from existing IFC Entities, Psets and Properties, or as shown below, can specify user-defined properties, which is a very popular feature in IFC for dealing with project-specific extensions. If a particular piece of information is required for a use case of interest, it can be

exported from BIMQ to support the setup of CAD authoring tools and create checking rules based on the open mvdXML format or other checking-tool-specific formats.

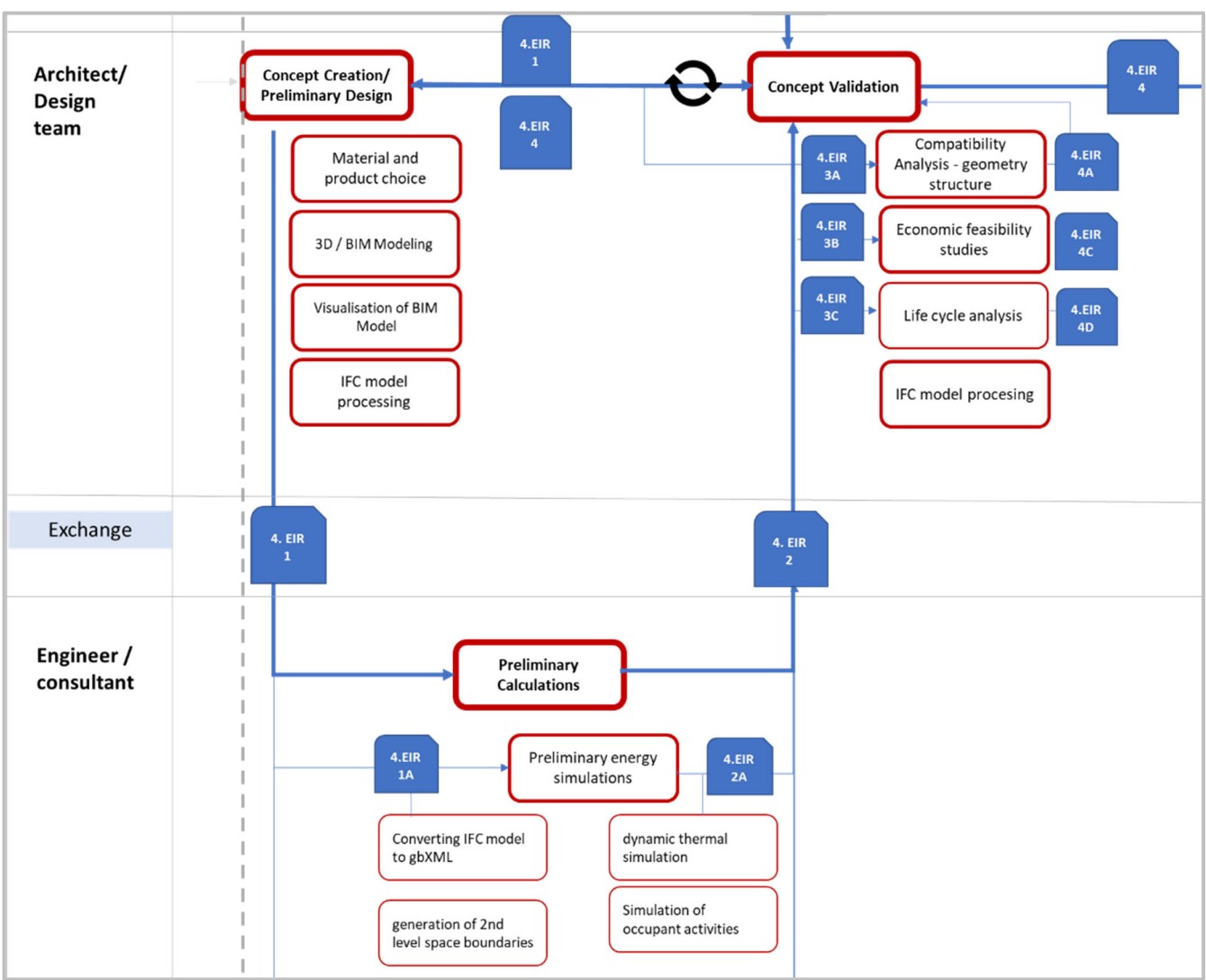

**Figure 10.** Process of the data exchange in the concept design phase.

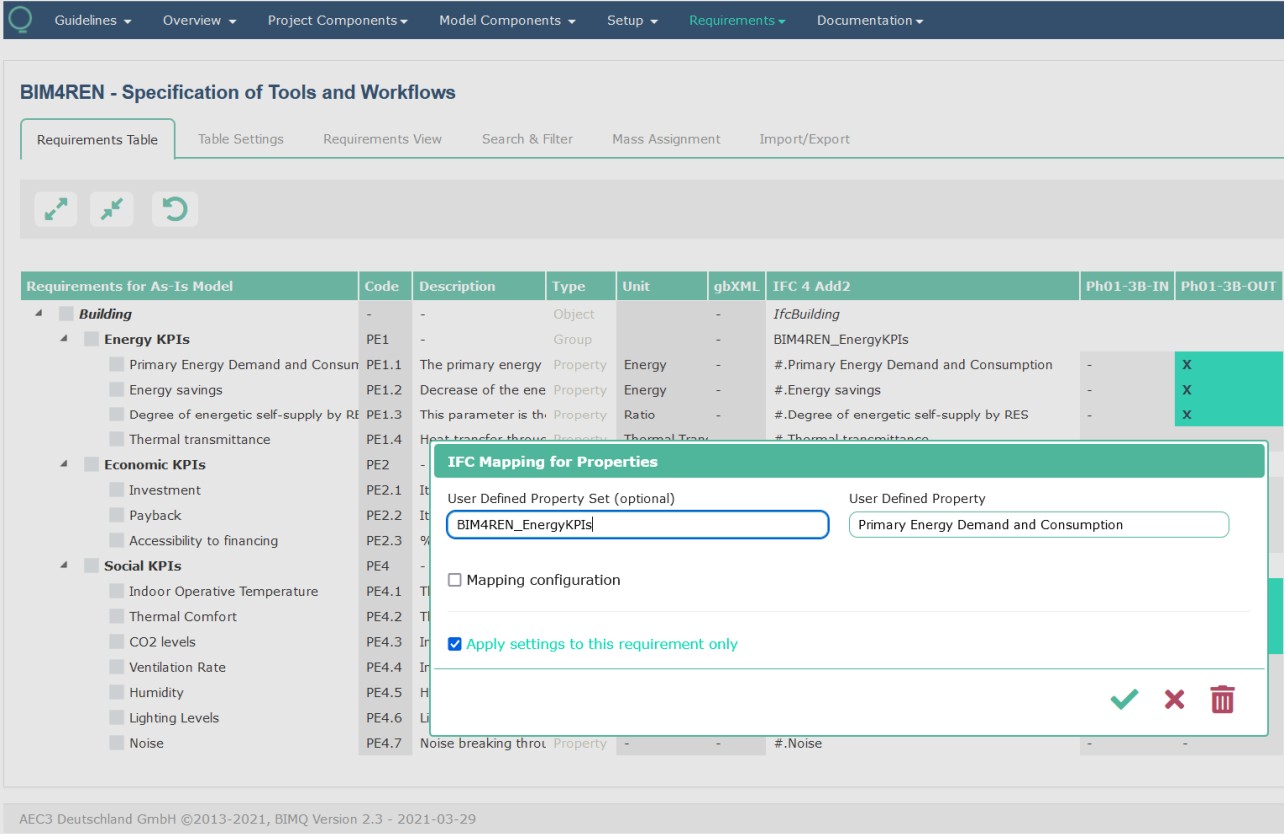

**Figure 11.** Support by BIMQ for organising EIRs and its IFC mapping definition.

## 4. Discussion and Conclusions

The only way to meet the challenge of improving productivity in construction for better profitability is through the integration of stakeholders, requirements, and technologies in workflows as a first precondition to automate the process. Although the methodology presented is valuable for the overall construction process, it is of the utmost importance in the case of renovation, owing to the clear need for holistic management of this complex process. The collaborative workflows for renovation play a crucial role in the integration of the phases and stakeholders around the digital model and they will pave the way for a smooth transition from a manual process to a leaner digital process.

Only suitable harmonization of terms and roles will allow one to set the basis for a common understanding, which leads to a collaborative environment for the necessary optimization of the process. In this sense, the current paper offers a classification of typologies, phases, and stakeholders involved in a renovation process as the first approach of this paramount standardization, which facilitates the design of renovation workflows. A huge amount of work is necessary to specify all details related to the identified EIRs. This requires further standardization efforts and the use of new tools such as BIMQ to not only support the specification, but also the reuse and sharing of such information. It is expected that this task is best accomplished by a constant refinement process, which means to evaluate and constantly adjust specifications. This also enables and fosters the use and development of new BIM-based services, because they can easily be integrated in the renovation workflow.

The involvement of more or new stakeholders with new insight and implications in the renovation, not considered in a general construction process, is a cornerstone in aretrofitting project. The integration of their expectations in the design and execution of

work must be understood as an opportunity to gain direct feedback and as a valuable collaboration source to ensure the project's success.

The standardization of the retrofitting process, taking into account the different perspectives and viewpoints of all stakeholders in organizational and technical (software) terms, will strengthen the efficient interchange of information and knowledge among the actors involved and the different software tools that are used in the process, rendering the information flows into a single window service, i.e., a leaner and more replicable faultless process. In this regard, and following the example in Section 3, the MVD/IFC View definition is eventually elaborated on top of the IDM to enforce software interoperability and enable the development of software tools that fulfill the requirements. In addition to achieving this technical interoperability, a collaboration-oriented culture must be naturally adopted to foster digital collaboration within an organizational frame. The role of a BIM manager belonging to each of the stakeholders will be the foundation to build solid organizational and, in turn, technical interoperability to root a seamless digital collaboration environment as conceived by the BIM paradigm.

In addition to the advantages of applying a systematic approach to the renovation process addressed in this paper using OpenBIM principles, e.g., boosting the renovation rate to reduce with carbon emissions according to the 2030 Climate Target Plan [3], the following capabilities are considered to be enhanced:

- Business Process measurement: the renovation process, since it is systematized by BPMN, can be measured and tracked with performance KPIs, since the different processes and actors are well identified, and the process is reproducible.
- Metrics analysis: The business processes can be benchmarked, and thus performance metrics can be obtained to determine bottlenecks and constraints, break them, and improve the process.
- BIM model generation: The collaborative approach driven by BIM managers and supported by the administration will lead to the generation of Digital Building Logbooks [75] or Digital Twins, which comprise the static data, stored in a geometric and semantic model, and dynamic data [76], generated by IoT sensors integrated in the built assets.
- Data exploitation: The digital logbooks, acting as single-data lakes that support the collaborative BIM process, will allow the exploitation of data through single APIs to apply simulations and calculations.

Moreover, since the last version of the data models is to be used by any of the actors to perform their activities, and the modifications are built upon the latest previous changes, the knowledge remains consolidated and structured in a way that potentially allows data analysis. These analytical capabilities can improve the process performance by application of machine learning technologies to support the operation, predictive maintenance, and end of life of the building stock in cooperation with existing legacy systems, such as Building Management Systems (BMS), Enterprise Resource Planning (ERPs) systems, and Computerized Maintenance Management Systems (CMMS).

The research presented in the current paper delivers, as a result, the methodology for the digitalization of renovation workflows showcased in a façade renovation use case, as one of the set of workflows developed in the pilots in the BIM4Ren project. The digital workflows themselves are also provided as companion material. As part of the future research, these digital workflows will support the definition of the BIM4Ren software system architecture and associated tools developed in the project, by means of further exploiting the MVD and IDS. The latter entails a novel mechanism to formalize information needs and its representation to describe the subset of the IFC schema needed to satisfy one or many exchange requirements of the AEC industry in the renovation process. These workflows, which will be revised and enhanced throughout the whole BIM4Ren project, will eventually incorporate the detailed EIR tables and associated model view definitions according to the approach described in this document and will enable an interoperable and

efficient renovation process by means of bringing together the involved stakeholders and software tools.

In this regard, the BIM4Ren project software tools will be adapted to cope with the digital workflows, ensuring that they meet the users' requirements. The workflows will help to digitally implement the renovation project, guiding the use of the software platform or tools, with the ability to support or execute the linked tasks to cover the whole renovation process. The methodology presented will enable an interoperable and efficient renovation process by means of bringing together the involved stakeholders and software tools.

The outcomes of this research will be further validated in the BIM4Ren pilots, where the optimization of the processes will be measured through KPIs, which will systematically allow the comparison of the manual process and the digital automated process performance in terms of efficiency. The measurement of performance in each of the process tasks under a continuous improvement approach, through systematized tasks, will permit the identification of process weak points as areas of improvement for process performance.

Moreover, future steps of this research will be focused on mapping the BIM software and tools to the BPMN diagrams to provide guidance to the end user to set up a renovation project according to the selected renovation scenario. It will allow the potential BIM tool developers to adjust the software functionalities strictly to the end users' requirements and orchestrate the tools under a collaborative and integrated perspective.

**Author Contributions:** Data curation, A.A., P.E., N.L. and M.W.; Investigation, A.A., P.E., N.L. and M.W.; Writing—original draft preparation, A.A., P.E. and N.L.; Writing—review & editing, A.A., P.E., N.L. and M.W. All authors have read and agreed to the published version of the manuscript.

**Funding:** This work has been developed within the project BIM4Ren. The project has received funding from the European Union's Horizon H2020 research and innovation program under Grant Agreement No 820773. This manuscript reflects only the author's views and the Commission is not responsible for any use that may be made of the information it contains.

**Conflicts of Interest:** The authors declare no conflict of interest.

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
