# Peer review of "A Methodology for the Digitalization of the Residential Building Renovation Process through OpenBIM-Based Workflows"

_applsci, doi:10.3390/app112110429_

Round 1

Reviewer 1 Report

Dear authors,
I would like to commend you for working in such important field as building renovation. As you write in your manuscript, it is indeed necessary to improve the efficiency of building renovations. The methodology you propose could help the stakeholders, however its presentation in the manuscript needs major improvements before publishing. Below are several comments I would like to ask you to address.

ENGLISH

I am not a native English speaker, but I still recommend one additional round of proofreading. Focus on inconsistent terminology and abbreviations. Also, some sentences are choppy or out of context in their current position (see Structure below). For example:

  • Line 358: Previous text mentioned two steps, but this paragraph starts with “First phase“. Is that correct? If so, I miss connection with previous text.
  • Line 361: What is “Use Case”? Is it the same as “Pilot site” mentioned before?
  • The manuscript introduces abbreviation EIR, but still uses full term Exchange Information Requirements in many occasions.
  • Lines 35-36: I recommend rephrasing the sentence. It is unclear, what it’s trying to tell.

STRUCTURE

The manuscript is fragmented. I recommend restructuring of the texts, expecially in sections 1 and 2 according to (at least) following comments:

  • The Introduction starts with description of the research and methodology. It should start if overview of the issues problems that lead to development of the methodology that are currently in Section 1.1. It could also contain details of the BIM4Ren project that are currently fragmented into sections 1.2, 2, …
  • Methodology should be described only in section 2. Its fragments are currently in sections 1 and 3.
  • The Introduction should end with clear establishment of a hypothesis/motivation/research goal justifying the need for this manuscript. This should be followed by a brief description of the manuscript’s content.
  • Text in section 1.1 mixes general information with outcomes of BIM4Ren project (under which this manuscript belongs) without sufficient background. Lack of references also doesn't improve clarity.
  • Section 1.2: Authors should clearly state, what is the relation of the 8 light listed issues and their methodology. Are all of them addressed? How and why?
  • Section 2.1: What’s the aim of this section? It’s title does not correspond with its content. Is it really necessary to list types of renovations and participants of the renovation work here? The text is rather general and poorly referenced. It mentions a literature review available in one of the deliverables of BIM4Ren project several times, but does not provide details.
  • Lines 376-453: Is it necessary to add such detailed description of the BPMN in this manuscript? I recommend shortening it and adding references to reliable sources.

REFERENCES
There's plenty of literature on building renovations, BIM, ... The manuscript mentions a throughout literature review that was performed under BIM4Ren project. Sadly, the manuscript itself is rather poorly referenced, which reduces its credibility. For example:

  • Lines 37-40: I miss references confirming the stated issues. Is the text describing your own experience or is it a result of literature review? Is it valid worldwide or only in specific regions?
  • Line 57 states that “… many literatures tackle…” but provide only one reference to a website. In fact, a lot of references are only websites, self-references or standards. I recommend adding more references to reliable sources (especially) to sections 1 and 2. This would enhance the background information, clarify and justify your decisions.
  • Line 94 mentions “the questionnaire”. From the context I understand that this questionnaire was crucial part of presented research. Yet, it is not referenced and it is only briefly mentioned in other parts of the manuscript.
  • Figure 1: A reference is missing (probably [27]).
  • Lines 166-216 include only one reference. Does all this information really come from this single reference?

METHODOLOGY AND ITS DESCRIPTION

The description of the methodology itself have to be improved. The title and introduction suggest that the methodology is intended for building renovations. However from the content of the manuscript it seems that it could be used for any construction. Add more details that highlight why it is useful specifically for renovations. For example, describe the diagrams of renovations that you developed in more details (e.g. reasoning, optimization, ...). Below are several further suggestions:

  • Lines 146-147: The pilot sites you mention should be specified by at least a geographical location. Brief description and reasoning behind their selection might also increase clarity of the manuscript.
  • Lines 150-152: Why is this paper based only on data from the “Spanish pilot”? Wouldn’t it reduce applicability of the methodology in other conditions/regions?
  • Section 2.3: It is unclear, why BPMN was selected.
  • Figure 1: Why was this particular diagram added? It describes package delivery… Text in lines 430-453 mentions that a set of diagrams was created during authors‘ work. It would be better to provide one of these in my opinion.
  • Replace the figures with High-Res versions in the manuscript.

Reviewer 2 Report

It is advisable to rethink the title. The paper refers more to organizational methodologies than to the use of BIM/OpenBIM technology in the rehabilitation process. As the research does not go deeper into the transfer of specific data between stakeholders regarding the use of BIM, the title and keywords lead to confusion.

In cases of retrofitting of residential buildings, the lack of presence of the residents in the whole process is striking. It is important to include them in the moments of prior information, explanation of technical and economic solutions, a timetable, and effects during the construction process. In fact, a key and important moment to be solved should focus on how to convey the technical solutions graphically to the residents. The example lacks photos of BIM plans of the façade to show how to use the graphic capacity of BIM to incorporate non-technical people into the process.

It is convenient to include the time variable in the whole rehabilitation process. It can be interesting to include in the example the time spent in each of the phases. If you want to control the economy and improve performance in any development, time is fundamental. A proposal to improve the BPMN is to try to include the time component in the development of the whole project. 

Line 526-527. Figure 9. Generic Façade renovation BPMN diagram (ID2) has not sufficient quality to be understood. It is proposed to improve its presentation to make the diagram readable.

Reviewer 3 Report

REVIEW: A Methodology for the digitalisation of the residential building renovation process through OpenBIM based workflows

This study aimed at improving the efficiency in building renovation processes by the use of OpenBIM technique. In OpenBIM, the information between multi-stakeholders, which is formulated as Exchange Information Requirements (EIR), can be exchanged using IDM (Information Delivery Manual) and the graphic language BPMN. The following questions and suggestions are provided:

S1. Introduction:

  • The authors introduced the existing barriers in the renovation process, and the requirements of applying BIM in renovation. How to ensure the reliability of the mentioned barrier factors only based on one BIM4Ren project? Why these barriers should be overcome? What are the advantages of solving these barriers using OpenBIM method? These questions should be clearly addressed for highlighting the academic and practical contributions (e.g., for enhancing the efficiency in renovation process using BIM method).

S2.1.  

  • Lines 173-174: The authors stated the renovation process was found from literature review and interview. However, the reliability of the research data source should be justified. For example, the detailed information about the literature, the quality of the literature, the information extracted from the literature, the profile of the interviewer, the detail of interviews, should be given.
  • Line 223: More references should be stated for proving the renovation classification.

S2.2 to S2.4.

  • The authors focused on introducing the roles, functions, and operating instructions of BIM, BPMN, and MVD. However, the main dish should be “using BIM to exchange information among different stakeholders at renovation stage”. This reviewer expects to have learn “what information should be used, how the information being exchanged” in this section.

 S3.

  • The authors used BIM4Ren project as the case study. More background information of the BIM4Ren case should be presented. The rationales of using this case for validation should also be given.
  • No solid conclusion can be made, in relation to the improved efficiency using OpenBIM. In other word, this reviewer has no clue how the contributions can be validated in practice settings.

In overall, although the authors claimed that the novelty of this research study is the information exchange using BIM technology, Unfortunately, this presented works focused too much on introducing the available OpenBIM tools which may not be necessary. The mechanism of exchanging the information was not well presented. The contributions were also not validated and verified. Given the fact that the envisioned amendment process and review process of this research article will be too time-consuming. The reviewer is not supportive to the paper publication at the current moment.

(*Decline)

Round 2

Reviewer 1 Report

The manuscript was significantly extended during the revisions. Sadly, there are still several issues requiring author's attention:

STRUCTURE, METHODOLOGY and IMPACT

The content and structure of the manuscript were notably improved. However the following issues remain unresolved in my opinion:

  • Lines 70-90: Section 1.1 should end with clear establishment of a hypothesis (BIM will improve efficiency of renovations?) and proposal of a solution (BIM4Ren project?). It is confusing to read about the aim of the manuscript in page 4! Last two paragraphs of section 1.1 are not specific enough. Moreover, barriers should be described before their solution. Therefore, these paragraphs should switch places.
  • Line 93: I suggest specifying the partners. How many partners are research institutions, how many are construction companies, stakeholders, etc.

  • Line 98: I recommend stating the pilot sites here.

  • Please, add information about dissemination of the project result to the manuscript. Will it be implemented in some legal documents or will it remain a voluntary supplement available at the project website? Do you have a feedback from stakeholders in this regard?

REFERENCES

  • Lines 57-60: The text implies that BIM is spreading in the whole construction sector worldwide. However, the references [18, 19] are both only about transport infrastructure. I recommend 1) adding more relevant references and 2) adding some information regarding validity of this statement (e.g. is it valid worldwide or only in developer countries, …).
  • References [32,33] are rather old. Aren’t there any newer references?
  • Line 211: Statement that something is “normally” (“commonly” is better word) known should be supported by references.

ENGLISH

The manuscript still contains a lot of grammar errors, typos and terminology uncertainties. It seems that the added parts were not proofread before submission. I have to request significant improvement of English. below are examples of errors I found in sections 1 and 2 only:

  • Line 27: „…IN the environment…“ should be corrected to „…ON the environment…“.
  • Line 35: „Lifecycle in building construction…“ could be corrected to „Life cycle of a building…“. Also, try to avoid emotional adjectives such as „huge“.
  • Line 40: „… being…“ should be corrected to „… making…“
  • Line 44: What do you mean by “building process”? The construction/renovation? Or the whole building life cycle?
  • Lines 42 – 47: Both sentences in this paragraph say almost the same thing. I recommend merging them together (and adding them to the previous/following paragraph).
  • Lines 48 – 52: Please, be specific and avoid synonyms (or would be synonyms) that might confuse the reader of the manuscript. An example is in this text. I presume, that “This source…” in line 52 refers to “… a design…” in line 51, but it is not clear.
  • Line 53-54: I suggest changing “The data exchanged…” to “The exchanged data include…” to improve the word order and clarity of the sentence.
  • Lines 70-71: I believe that “… barriers that difficult the flow…” is an incomplete statement.
  • Line 75: I suggest correcting “high potentiality” to something like “high potential”. Also, please correct the word order in the sentence.
  • Line 91: Is the heading “Background. BIM4Ren Project” correct? Why is there a dot after “Background”?
  • Lines 95-96: Please correct/ rephrase “…are been developed oriented to…”. It is unclear what this statement means.
  • Line 97: What do you mean by “developments”?
  • Line 101: Abbreviation LL (Living Labs) is introduced. However it is not applied thoroughly in the following texts.
  • Line 116: Please, correct „… the basis to systemitize the…“ for example to „… the basis for systemization of…“
  • Lines 109-122: This paragraph is confusing due to overuse of synonyms. It starts with definition of „final goal“, which is later replaced by „final stage“. It states that the project comprises of three steps, but later mentions „initial stage“, which should probably be a synonym for „first step“. Later, in section 2 there are also “steps”, which are different than those mentioned here. Also, the boundary between the steps is unclear. I recommend rephrasing the whole paragraph (and the following texts) to make the manuscript more comprehensive.
  • Line 126: In relation to the previous comment, what do you mean by “final stage” in this line? Is it the final stage of the whole project or just the final stage of the first step?
  • Line 131: Correct the start of the sentence “Two are the main…”.
  • Line 132: What do you mean by “… implementation of the main actors…”?
  • Line 134: “… connected to…” should be corrected to “… connected with…”.
  • Section 1.3: Is the title accurate? Is this section really covering whole BIM4Ren or just the content of this paper? If the former than it should be probably merged with section 1.2. If the latter than it should be probably merged with the end of section 1.1.
  • Line 153: Please, rephrase “This implies to apply…”.
  • Line 159: Please, rephrase “… is useful to be used…”

Reviewer 3 Report

This paper is revised positively and sincerely. This reviewer is so happy that the authors addressed all my comments meticulously. As such, given the improved manuscript, the reviewer supports the publication this time. Lastly, the reviewers suggest the authors make references to (cite) the following articles which are highly related to this article.

  1. Chan, D.W.M.; Cristofaro, M.; Nassereddine, H.; Yiu, N.S.N.; Sarvari, H. Perceptions of Safety Climate in Construction Projects between Workers and Managers/Supervisors in the Developing Country of Iran. Sustainability 2021, 13, 10398. https://doi.org/10.3390/su131810398
  2. Saka, A. B., Chan, W. M., and Siu, M. F. (2020) “Drivers of Sustainable Adoption of Building Information Modelling (BIM) in the Nigerian Construction Small and Medium-Sized Enterprises (SMEs).” Sustainability.
  3. Saka, A., Chan, W. M., and Siu, M. F. (2019). “Adoption of Building Information Modelling in Small and Medium-Sized Enterprises in Developing Countries: A System Dynamics Approach.” CIB Conference 2019. Honour: Best Student Presentation Award (Saka, A).

Congratulations, and hope to read the published paper once available.
